# Comparison of four low-cost carbapenemase detection tests and a proposal of an algorithm for early detection of carbapenemase-producing *Enterobacteriaceae* in resource-limited settings

**Wirittamulla Gamage Maheshika Kumudunie[1], Lakmini Inoka Wijesooriya[2], Yasanandana Supunsiri Wijayasinghe[1]***

**1** Department of Biochemistry and Clinical Chemistry, Faculty of Medicine, University of Kelaniya, Ragama, Sri Lanka, **2** Department of Medical Microbiology, Faculty of Medicine, University of Kelaniya, Ragama, Sri Lanka

* supunw@kln.ac.lk

**Data Availability Statement:** All relevant data are within the manuscript.

## Abstract

Rapidly progressing antibiotic resistance is a great challenge in therapy. In particular, the infections caused by carbapenem-resistant *Enterobacteriaceae* (CRE) are exceedingly difficult to treat. Carbapenemase production is the predominant mechanism of resistance in CRE. Early and accurate identification of carbapenemase-producing carbapenem-resistant *Enterobacteriaceae* (CP-CRE) is extremely important for the treatment and prevention of such infections. In the present study, four phenotypic carbapenemase detection tests were compared and an algorithm was developed for rapid and cost-effective identification of CP-CRE. A total of 117 *Enterobacteriaceae* (54 CP-CRE, 3 non-CP-CRE, and 60 non-CRE) isolates were tested for carbapenemase production using modified Hodge test (MHT), modified carbapenem inactivation method (mCIM), Carba NP test (CNPt), and CNPt-direct test. The overall sensitivity/specificity values were 90.7%/92.1% for MHT, 100%/100% for mCIM, 75.9%/100% for CNPt, and 83.3%/100% for CNPt-direct. OXA-48-like enzymes were detected with 93.2% sensitivity by MHT and >77.3% sensitivity by two Carba NP tests. MHT could only detect half of the NDM carbapenemase producers. CNPt-direct exhibited enhanced sensitivity compared to CNPt (100% vs 25%) for detection of NDM producers. Considering these findings we propose CNPt-direct as the first test followed by mCIM for rapid detection of CP-CRE. With this algorithm >80% of the CP-CRE could be detected within 24 hours from the time the sample is received and 100% CP-CRE could be detected in day two. In conclusion, mCIM was the most sensitive assay for the identification of CP-CRE. CNPt-direct performed better than CNPt. An algorithm consisting CNPt-direct and mCIM allows rapid and reliable detection of carbapenemase production in resource-limited settings.

**Funding:** This study was supported by a research grant to YSW (Grant No. NRC 17-055) from the National Research Council of Sri Lanka. The funders had no role in study design, data collection and analysis, decision to publish, or preparation of the manuscript.

**Competing interests:** The authors have declared that no competing interests exist.

## Introduction

Carbapenems are the antibiotics of "last-resort" that are used to treat severe infections caused by drug-resistant *Enterobacteriaceae* such as *Klebsiella* spp., *Escherichia* spp., *Enterobacter* spp., etc., which are frequently associated with both nosocomial and community-acquired infections. The emergence of carbapenem resistance in *Enterobacteriaceae* (CRE) poses a serious public health threat as these infections are difficult to treat [1].

Carbapenem resistance is caused by a number of mechanisms, but mainly by the production of carbapenem hydrolyzing enzymes (carbapenemases) [2]. Carbapenemases are of several types and are grouped into three β-lactamase Ambler classes: class A (KPC is the predominant), class B (includes metallo β–lactamases; MBLs, eg: NDM, VIM, IMP), and class D (eg: OXA-48-like). It has been found that the infections caused by carbapenemase-producing carbapenem-resistant *Enterobacteriaceae* (CP-CRE) are associated with greater threat to human health with high fatality rates and increased healthcare cost, compared to the infections caused by non-CP-CRE [3–6]. Carbapenemase-producing microorganisms have a high potential to cause outbreaks as carbapenemase genes can be easily transferred to the sensitive bacteria [7–10]. CP-CRE are also less likely to be susceptible to other groups of antibiotics, such as aminoglycosides, fluoroquinolones, and polymyxins [4]. In addition, distinguishing carbapenemase producers from non-carbapenemase producers is important to prioritize newer antibiotics for the infection caused by CP-CRE, which might delay the development of resistance to novel drugs. Thus, rapid and reliable detection of CP-CRE is becoming increasingly important to streamline antibiotic treatment and also to minimize the spread of these difficult-to-treat bacteria both in healthcare facilities and in the community.

Several laboratory methods are available for the identification of CP-CRE. Molecular detection by polymerase chain reactions (PCR) is considered as the gold standard for the confirmation of carbapenemase production [11]. However due to high cost, unavailability of necessary instrumentation and technical expertise in many diagnostic laboratories, the PCR is not suitable for routine detection of CP-CRE in developing countries. On the other hand, PCR detects only the genes encoding the known enzymes and hence the organisms harbor novel carbapenemase genes are possibly undetected [12]. In recent years, a number of simple phenotypic methods have been developed for the identification of carbapenemase production, including colorimetric biochemical assays (eg: Carbapenemase Nordmann-Poirel or Carba NP test) and growth based carbapenem inactivation disc assays (eg: modified Hodge test (MHT), modified carbapenem inactivation method) [13].

At present, Carba NP test (CNPt) and modified carbapenem inactivation method (mCIM) are the only carbapenemase confirmatory tests endorsed by the Clinical and Laboratory Standard Institute (CLSI) [14, 15]. However, there is no perfect method to identify all CP-CRE as different tests show variable sensitivities and specificities depending on the carbapenemase type [16, 17]. Therefore, it is recommended to select a carbapenemase detection test based on the molecular epidemiology of CP-CRE in a given geographical area. In addition, user-friendliness, cost effectiveness, and turnaround time should also be considered when deciding which method to be employed in a clinical diagnostic laboratory [18].

In the present study, we assessed the performance of four phenotypic carbapenemase detection tests, namely MHT, mCIM, and CNPt and Carba NP-direct (CNPt-direct). CNPt-direct is a recently developed variant of Carba NP assay, in which the proprietary bacterial lysis solution (B-PER II) is replaced with Triton-X100 (a nonionic detergent) to reduce the cost of CNPt. In addition, CNPt-direct has also demonstrated superior sensitivity in detecting CP-CRE [19]. All four tests are based on the ability of carbapenemases to hydrolyze carbapenems. These tests are inexpensive compared to the PCR based detection methods, hence can

be easily performed at routine clinical microbiology laboratories in developing countries. Moreover, we also proposed an algorithm for the rapid identification of CP-CRE in resource-limited settings.

## Materials and methods

### Bacterial isolates

A total of 117 previously characterized *Enterobacteriaceae* clinical isolates were tested to evaluate the performance of four phenotypic carbapenemase detection assays, namely MHT, mCIM, CNPt, and CNPt-direct [20]. These strains included 54 genetically identified CP-CRE producing carbapenemases of Ambler class A. B, and D: KPC (n = 2), NDM (n = 4), OXA-48-like (n = 44), NDM+OXA-48-like (n = 4). Three non-CP-CRE and 60 non-CRE consisted of 30 extended-spectrum β-lactamase producing *Enterobacteriaceae* (ESBL-PE: ESBL production was identified by double disc synergy test using 30 μg cefotaxime and 20/10 μg amoxicillin-clavulanic acid discs according to CLSI guidelines) and 30 non-ESBL-PE (*Enterobacteriaceae* sensitive to both third generation cephalosporins and carbapenems) served as negative control in the study. *K. pneumoniae* ATCC BAA-1705 possessing KPC, *K. pneumoniae* ATCC BAA-2146 and *E. coli* ATCC BAA-2469 possessing NDM-1 and *K. pneumoniae* ATCC BAA-2524 possessing OXA-48-like carbapenemases were used as positive controls. *K. pneumoniae* ATCC BAA-1706 strain was used as the negative control.

### Modified Hodge test

The MHT was performed and interpreted according to CLSI M100-Ed27 guidelines [15]. Briefly, 10 μg ertapenem (Oxoid, UK) disc was placed on the center of Mueller-Hinton agar (MHA) (Oxoid, UK) plate which had been previously inoculated with an *E.coli* indicator strain (ATCC 25922). The suspension of the indicator strain was prepared by emulsifying bacterial colonies in 5 ml normal saline and adjusting the turbidity similar to 0.5 McFarland standard solution. The straight lines of positive control organism, negative control organism and test organism were streaked from the edge of the disc to the edge of the plate and subsequently incubated at 37˚C for 18–24 hours. The appearance of clover-leaf like indentation of the indicator strain growing along the streaks of the bacterial strain was indicative of carbapenemase production.

### Modified carbapenem inactivation method

The mCIM was performed and interpreted according to CLSI M100-Ed28 guidelines [21]. Briefly, a 1 μL loopful of overnight grown bacteria on blood agar was emulsified in 2 mL of Tryptic Soy Broth (TSB) (Oxoid, UK). In parallel, a second 2 mL TSB tube was labeled for each isolate for EDTA modified carbapenem inactivation method (eCIM) test in with 0.5 mM EDTA (pH 8.0). Each tube was vortexed for 10–15 seconds. Subsequently, a 10 μg meropenem disk (Oxoid, UK) was immersed in each bacterial suspension and incubated at 37˚C. After 4 hours of incubation, meropenem disk in each bacterial suspension was placed on MHA (Oxoid, UK) plates which had been freshly inoculated with the 0.5 McFarland dilution of a susceptible *E. coli* indicator strain (ATCC 25922). The plates were incubated at 37˚C for 18–24 h. Following day, the zone diameters were measured and interpreted according to the established CLSI standards. An inhibition zone diameter of ≤ 15 mm was indicative of CP-CRE. Besides, a ≥ 5 mm increase in zone diameter of eCIM compared to that of mCIM considered the presence of metallo β-lactamase type (i.e. Ambler class B) carbapenemases.

## Carba NP test

CNPt was performed as per CLSI M100-Ed28 guidelines with slight modifications [21], briefly bacterial lysate was prepared in a 1.5 ml micro centrifuge tube by suspending a 10 μl loopful of overnight grown bacterial colonies on MHA (Oxoid, UK) or Blood Agar (Oxoid, UK) in 100 μl of B-PER II bacterial lysis solution (Thermo Scientific, USA) followed by vortexing and incubation for 15 minutes at room temperature. CLSI recommend using 1 μl loopful of bacteria, however, it was reported that more bacteria could increase the test sensitivity [22]. Then it was mixed with 100 μl of an indicator solution (Solution A) which had previously been adjusted to pH 7.8, consisting phenol red (0.005% W/V), 0.1 mM $ZnSO_4$ and 6 mg/mL imipenem monohydrate (Sigma-Aldrich, USA) or 12 mg/mL imipenem-cilastatin (Tienam 500, Merck, USA). Then the mixture of the enzymatic suspension being tested and the revealing solution was incubated at 37˚C up to 2 hours. The observations for the color change of phenol red were recorded in every 15 minutes up to 2 hours. The results were interpreted according to CLSI guidelines. The change of the color of reaction medium from red to yellow or to light orange was considered as positive for carbapenemase production.

## Carba NP-direct test

CNPt-direct test was performed as previously described [19]. Briefly, the indicator solution (Solution A) consisting phenol red (0.005% W/V) and 0.1 mM $ZnSO_4$ was mixed with 0.1% of Triton X-100 (Sigma Aldrich) and pH was adjusted to 7.8 with 1N NaOH. A 10 μl loopful of pure bacterial colonies was directly suspended in 1.5 ml micro centrifuge tubes containing 100 μl of CNPt-direct mix supplemented with 6 mg/mL imipenem monohydrate or 12 mg/mL imipenem-cilastatin (Tienam 500, Merck, USA), the pharmaceutical form of imipenem for test reactions and without antibiotic for control reactions. Finally, all the tubes were incubated at 37˚C and were monitored every 15 minutes up to 2 hours. Color change from red to yellow or to light orange was considered as positives for carbapenemase production. The pH adjusted, imipenem unsupplemented CNPt and CNPt-direct solutions were stored at 4˚C and used within 4 weeks. The test solution was supplemented with imipenem or imipenem-cilastatin immediately before use.

## Statistics

Sensitivity, specificity, positive and, negative predictive values, and associated confidence intervals (CI) were calculated for each test using MedCalc, an internet-based statistical software (Available from: https://www.medcalc.org/calc/diagnostic_test.php). The prevalence of CP-CRE was considered as 10% in the calculation of positive and negative predictive values. The sensitivities were compared by McNemar's test. *P* values <0.05 were considered statistically significant.

## Results

### Overall performance of four carbapenemase detection tests

All four tests exhibited above 75% sensitivity for the overall detection of carbapenemase production in *Enterobacteriaceae* clinical isolates. The performance characteristics of each assay are presented in Table 1. Among four tests, only mCIM showed 100% sensitivity, specificity, positive and negative predictive values in detecting carbapenemase producers. Of 54 CP-CRE, 49 were identified by MHT (sensitivity of 90.7%). Five *Enterobacteriaceae* in negative control group (i.e. non-CP-CRE and non-CRE) produced false-positive results in MHT yielding specificity of 92.1%. MHT had negative predictive value of 98.9% and positive predictive value of

**Table 1. Sensitivity, specificity, positive and negative predictive values of four carbapenemase detection tests.**

| *Enterobacteriaceae* isolate | Carbapenemase detection test | | | |
|---|---|---|---|---|
| | MHT n (%) | mCIM n (%) | CNPt n (%) | CNPt-direct n (%) |
| CP-CRE N = 54 | 49 (90.7%) | 54 (100%) | 41 (75.9%) | 45 (83.3%) |
| Non-CP-CRE N = 3 | 1 (33.3%) | 0 | 0 | 0 |
| Non-CRE N = 60 | 4 (6.7%) | 0 | 0 | 0 |
| Sensitivity, % (95% CI) | 90.7% (79.7–96.9) | 100% (93.4–100) | 75.9% (62.4–86.5) | 83.3% (70.7–92.1) |
| Specificity, % (95% CI) | 92.1% (82.4–97.4) | 100% (94.3–100) | 100% (94.3–100) | 100% (94.3–100) |
| Positive predictive value, % (95% CI) | 56.0% (35.3–74.7) | 100% | 100% | 100% |
| Negative predictive value, % (95% CI) | 98.9% (97.5–99.5) | 100% | 97.4% (95.9–98.4) | 98.2% (96.8–99.0) |

56.0%. All non-CP-CRE and non CRE were identified as negative by CNPt and CNPt-direct resulting in 100% specificity and positive predictive values. The sensitivity and negative predictive value were 75.9% and 97.4%, respectively for CNPt, and those were increased to 83.3% and 98.2%, respectively when tested with CNPt-direct. However, CNPt and CNPt-direct were not significantly different with respect to sensitivity ($P = 0.125$). The pharmaceutical imipenem (i.e. imipenem-cilastatin) generated identical results to authentic imipenem (i.e. imipenem monohydrate) in both CNPt and CNPt-direct.

## Enzyme type and organism-wise performance of carbapenemase detection tests

The performance of MHT and CNP tests varied among different carbapenemase types and the organism. The sensitivities of the four assays for detecting different carbapenemase types in different *Enterobacteriaceae* species are shown in Table 2. The MHT had lower sensitivities for the detection of NDM (50%) and OXA-48-like (93.2%) producers. CNPt-direct performed better than CNPt for detection of NDM carbapenemase (sensitivities 100% vs 25%), but their sensitivity for the detection of OXA-48-like producers ranged from 77 to 80%. The single NDM producing *Enterobacter cloacae*, *Providencia rettgeri*, and one out of two *Citrobacter freundii* isolates produced false-negative results in CNPt. While, all four NDM producing *Enterobacteriaceae* were detected by CNPt-direct. Only 31 of 39 (79.5%) OXA-48-like producing *K. pneumoniae* and 2 of 3 (66.7%) OXA-48-like producing *E. coli* isolates were detected by both CNPt and CNPt-direct. Though, only one out of two OXA-48-like producing *C. freundii* isolate was

**Table 2. Performance of four carbapenemase detection tests according to the type of carbapenemase and organism.**

| Carbapenemase type | *Enterobacteriaceae* species (n) | MHT n (%) | mCIM n (%) | eCIM n (%) | CNPt n (%) | CNPt-direct n (%) |
|---|---|---|---|---|---|---|
| OXA-48-like N = 44 | | 41 (93.2%) | 44 (100%) | 4 (9.1%) | 34 (77.3%) | 35 (79.5%) |
| | *K. pneumoniae* (39) | 37 (94.9%) | 39 (100%) | 4 (10.3%) | 31 (79.5%) | 31 (79.5%) |
| | *E. coli* (3) | 2 (66.7%) | 3 (100%) | 0 | 2 (66.7%) | 2 (66.7%) |
| | *C. freundii* (2) | 2 (100%) | 2 (100%) | 0 | 1 (50%) | 2 (100%) |
| NDM N = 04 | | 2 (50%) | 4 (100%) | 4 (100%) | 1 (25%) | 4 (100%) |
| | *C. freundii* (2) | 1 (50%) | 2 (100%) | 2 (100%) | 1 (50%) | 2 (100%) |
| | *P. rettgeri* (1) | 0 | 1 (100%) | 1 (100%) | 0 | 1 (100%) |
| | *E. cloacae* (1) | 1 (100%) | 1 (100%) | 1 (100%) | 0 | 1 (100%) |
| OXA-48-like & NDM N = 04 | | 4 (100%) | 4 (100%) | 0 | 4 (100%) | 4 (100%) |
| | *K. pneumoniae* (3) | 3 (100%) | 3 (100%) | 0 | 3 (100%) | 3 (100%) |
| | *K. aerogenes* (1) | 1 (100%) | 1 (100%) | 0 | 1 (100%) | 1 (100%) |
| KPC N = 02 | *K. pneumoniae* (2) | 2 (100%) | 2 (100%) | 0 | 2 (100%) | 2 (100%) |

identified by CNPt, both isolates were identified by CNPt-direct. All KPC producers and NDM + OXA-48-like coproduces were identified by all tests.

To assess the diagnostic value of EDTA modified carbapenem inactivation method (eCIM) in the detection of MBL producers (i.e. NDM in this study), we analyzed 57 CRE. eCIM detected all 4 NDM producers with 100% sensitivity. However, four (10.3%) OXA-48-like producing *K. pneumoniae* isolates yielded false-positive results.

## Discussion

Emergence and world-wide dissemination of CP-CRE is a major public health concern [23]. Rapid and accurate detection of CP-CRE is important as early identification can optimize antibiotic therapy and minimize unnecessary/ inappropriate prescription of medicines [24]. Although the conventional antimicrobial susceptibility testing is sufficient for the selection of appropriate antibiotics, commonly used disc diffusion method has a minimum of two day delay to results and is also not suitable for some antibiotics like colistin. Moreover, CLSI recommended breakpoints are not available for a number of antibiotics such as tigecycline, and for newer antibiotics [21, 25]. Therefore, timely identification of CP-CRE is important to guide the antibiotic treatment decisions and to minimize the spread.

Modified Hodge test is the first CP-CRE confirmatory method recommended by CLSI. Although MHT has been excluded from current CLSI guidelines due to its poor performance, it is still being used in developing countries to detect CP-CRE [26, 27]. In the present study, we observed relatively low overall sensitivity (90.7%) and specificity (92.1%), and low positive predictive value (56.0%) for MHT (Table 1). The poor sensitivity of MHT towards the identification of NDM producing CRE greatly contributed to the observed low positive predictive value of MHT (Table 2). Although, this study consisted of a small sample of NDM producing CRE (n = 4), these findings are in agreement with previous studies in which MHT has been found to have poor sensitivity (50–86%) for detection of MBLs [18, 28] and low specificity (91%) [18]. False positive MHT results have been reported with ESBL producers (Mostly CTX-M type and AmpC) with reduced membrane permeability [28, 29]. However, MHT has been found to have excellent sensitivity for detecting Ambler classes A and D carbapenemases [18, 28]. Since MHT is a simple, inexpensive test, and detect all KPC, all OXA-48+NDM, and above 90% of OXA-48-like producers, it can still be used in the regions, where KPC and/or OXA-48-like carbapenemases are predominant. However, it is worth to note that NDM is now wide spread and is growing. On the other hand interpretation of MHT is subjective and has a long turnaround time.

In the present study, mCIM demonstrated to be an excellent phenotypic test for carbapenemase detection (Table 1). Our findings corroborate the results of previous studies that have reported nearly 100% sensitivity and specificity of mCIM for detecting commonly found carbapenemase types such as KPC, NDM, VIM, IMP, and OXA-48-like [18, 30, 31]. mCIM is suitable for a resource-poor microbiology laboratory as it is inexpensive and easy to perform. Interpretations of mCIM results were less subjective than that of MHT. However, a major drawback of this method is that it has a longer turnaround time as it needs an overnight incubation. It is important to distinguish KPC and OXA-48-like (Ambler class A and D) from NDM (class B) enzyme producers as some of the new antibiotics combinations such as ceftazidime-avibactam are effective only against KPC and OXA-48-like (i.e. serine carbapenemase) producing CRE [32, 33]. Since, metallo-β-lactamases are inhibited by EDTA, EDTA-modified carbapenem inactivation method has been recommended by CLSI for the detection of MBL producing *Enterobacteriaceae* [21]. eCIM can be performed side-by-side with mCIM in the same culture plate. Hence, we also evaluated the performance of eCIM to detect MBL

producing CRE and found 100% sensitivity for the detection of NDM producers. As expected OXA-48-like+NDM co-producers did not yield positive results in eCIM, due to the fact that OXA-48-like enzymes are not inhibited by EDTA. Surprisingly, approximately 10% of OXA-48-like producers were also positive for eCIM (Table 2). eCIM has been reported to have 100% sensitivity and specificity for detecting MBLs [34]. However, eCIM was recently showed to produce false positive results with OXA-48-like enzymes [35, 36]. eCIM was also found to be less sensitive for the detection of IMP type MBLs (sensitivity 79.6%), but it was 100% sensitive for NDM enzymes [36]. Therefore, further studies with large samples are required to validate the performance of eCIM.

Routine culture-based antimicrobial susceptibility testing takes 48 hours or more from sample collection to reporting of results. It takes another 24 hours to test antibiotics that are not frequently used. Therefore, rapid diagnostic tests can shorten the turnaround time and hence the time to appropriate therapy. In that regard, CNPt is a valuable test for the detection of carbapenemase activity as it takes less than two hours to produce results. CNPt is a biochemical test based on *in vitro* hydrolysis of imipenem β-lactam ring resulting in a decrease of the pH of the medium, which is detected by the change of phenol red indicator color from red to yellow. CNPt has been claimed to provide rapid and accurate results for screening of CP-CRE [37].

In the current study, the overall sensitivity of both CNPt (75.9%) and CNPt-direct (83.3%) was low compared to that of MHT (90.7%). Both Carba NP tests had superior specificity (100%) than the specificity of MHT (92.1%) (Table 1). The original publication by Nordmann *et al.* has reported that CNPt was 100% sensitive and specific for CP-CRE [37]. Subsequent studies have reported lower (<90%) overall sensitivity of CNPt [22, 38, 39]. Tamma *et al.* has reported low sensitivity of manual CNPt (CLSI) for the detection of KPC (84%), NDM (92%), and OXA-48-like (40%) enzyme producing *Enterobacteriaceae* [18]. Further, CNPt has been repeatedly found less sensitive for OXA-48-like carbapenemases, as these enzymes have weak carbapenemase activity compared to the other enzymes [22, 38–40]. Our results were concordant with this observation as both CNPt and CNPt-direct showed low sensitivity (77.3% and 79.5%, respectively) for the detection of OXA-48-like carbapenemase in *Enterobacteriaceae* (Table 2).

Furthermore, the performance of CNPt-direct was better than CNPt for the detection of NDM CP-CRE as CNPt-direct detected all NDM producers, while CNPt failed to identify three out of four NDM isolates (Table 2). Therefore, the overall low sensitivity of CNPt-direct is mainly due to the false negative results of OXA-48 producers. However, due to the limited number of NDM used in this study, this observation needs to be further tested using a large group of NDM producers. It has also been reported that the NDM-1 producing *Providencia* spp with mucoid colonies tend to produce false-negative results [22]. CNPt has been reported to have lower sensitivity for isolates with mucoid colonies irrespective of the type of enzyme (eg: NDM-producing *P. rettgeri*, OXA-48-like-producing *K. pneumoniae* isolates) [22, 38, 41]. Tijet *et al.* has revealed that the false negative results obtained for mucoid metallo β-lactamase producing *Enterobacteriaceae* isolates were due to incomplete lysis of bacterial cell [40]. Pasteran *et al.* showed a higher sensitivity of CNPt-direct than CNPt (100% vs 71%) for the detection of NDM producing CRE [19]. The absence of a buffer in the lysis solution and efficient cell lysis by Triton X-100 are possible reasons for the superior performance of CNPt-direct. Carba NP tests detect the pH change when imipenem is hydrolyzed by carbapenemase. CNPt-direct simplifies the bacterial lysis step and shorten the time to color change in the test. However, the detection of OXA-48-like was not improved with CNPt-direct. Trton X-100 and imipenem-cilastatin are significantly cheaper than B-Per II and imipenem monohydrate, respectively. Therefore, the cost per CNPt-direct test is lower than that of CNPt.

Based on the findings of the present study, we propose an algorithm for the rapid detection of CP-CRE in resource-limited settings (Fig 1). In this algorithm, CNPt-direct is used as the

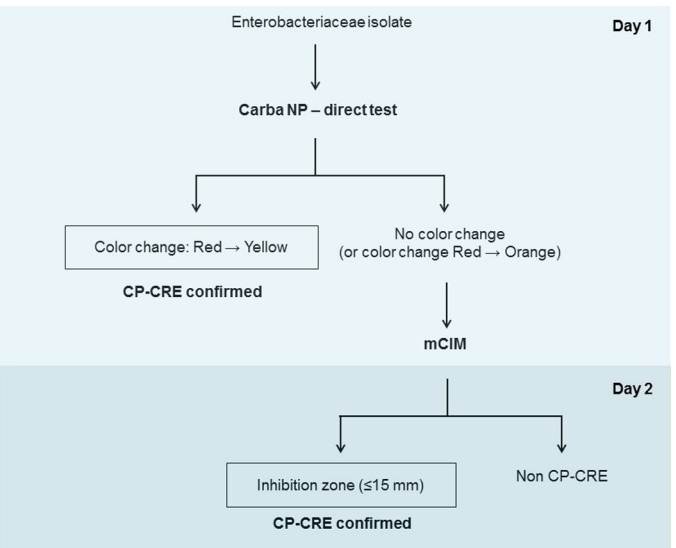

**Fig 1. Proposed algorithm for rapid detection of CP-CRE in resource-limited settings.**

first test, which can detect CP-CRE within 2 hours after isolation of bacterial colonies. The positive test confirmed the presence of carbapenemase producers. If negative, mCIM is performed as the second test. The positive mCIM test confirmed the presence of carbapenemases in *Enterobacteriaceae*. If mCIM is negative, the presence of CP-CRE is unlikely. In combination of CNPt-direct and mCIM, above 80% of CP-CRE were identified in day one and all (100%) CP-CRE were identified by day 2. The proposed algorithm is rapid, simple and affordable. This algorithm does not require specialized equipment or skilled personnel and hence it could be easily implemented in most of the clinical laboratories in resource-limited settings. However, this algorithm does not invalidate the importance of antimicrobial susceptibility test, which is required to guide antibiotic selection. Since the present study is mainly based on the OXA-48-like carbapenemase producing CRE, further studies with a large sample of CP-CRE isolates with adequate representation of commonly found genotypes (eg: KPC, NDM, OXA-48-like, IMP, VIM, etc.) may help better understanding of the performance of the proposed algorithm.

## Conclusions

mCIM was superior to Carba NP tests and MHT for the detection of CP-CRE. mCIM together with CNPt-direct offers a reliable, simple, rapid, and cost effective phenotypic carbapenemase detection algorithm that can routinely be employed in the clinical diagnostic laboratories in developing countries.

## Acknowledgments

We are thankful to the laboratory staff at the Department of Medical Microbiology, Faculty of Medicine, University of Kelaniya, Sri Lanka.

## Author Contributions

**Conceptualization:** Lakmini Inoka Wijesooriya, Yasanandana Supunsiri Wijayasinghe.

**Data curation:** Wirittamulla Gamage Maheshika Kumudunie, Lakmini Inoka Wijesooriya, Yasanandana Supunsiri Wijayasinghe.

**Funding acquisition:** Yasanandana Supunsiri Wijayasinghe.

**Investigation:** Wirittamulla Gamage Maheshika Kumudunie, Lakmini Inoka Wijesooriya, Yasanandana Supunsiri Wijayasinghe.

**Methodology:** Lakmini Inoka Wijesooriya, Yasanandana Supunsiri Wijayasinghe.

**Project administration:** Yasanandana Supunsiri Wijayasinghe.

**Supervision:** Lakmini Inoka Wijesooriya, Yasanandana Supunsiri Wijayasinghe.

**Validation:** Lakmini Inoka Wijesooriya, Yasanandana Supunsiri Wijayasinghe.

**Visualization:** Wirittamulla Gamage Maheshika Kumudunie, Yasanandana Supunsiri Wijayasinghe.

**Writing – original draft:** Wirittamulla Gamage Maheshika Kumudunie, Yasanandana Supunsiri Wijayasinghe.

**Writing – review & editing:** Wirittamulla Gamage Maheshika Kumudunie, Lakmini Inoka Wijesooriya, Yasanandana Supunsiri Wijayasinghe.

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
