## [Decision Letter · Decision Letter 0]

20 Nov 2020

PONE-D-20-31489

Comparison of four low-cost carbapenemase detection tests and a proposal of an algorithm for early detection of carbapenemase-producing Enterobacteriaceae in resource-limited settings.

PLOS ONE

Dear Dr. Wijayasinghe,

Thank you for submitting your manuscript to PLOS ONE. After careful consideration, we feel that it has merit but does not fully meet PLOS ONE’s publication criteria as it currently stands. Therefore, we invite you to submit a revised version of the manuscript that addresses the points raised during the review process.

The manuscript requires substantial revision in all sections

We look forward to receiving your revised manuscript.

Kind regards,

Iddya Karunasagar

Academic Editor

PLOS ONE

Journal Requirements:

2. Please provide more information on how another researcher could obtain the CRE strains used in this study so that they could reproduce your results.

3. To comply with PLOS ONE submission guidelines, in your Methods section, please provide additional information regarding your statistical analyses. In addition, please report your p-values to support your claims. For more information on PLOS ONE's expectations for statistical reporting, please see https://journals.plos.org/plosone/s/submission-guidelines.#loc-statistical-reporting.

Additional Editor Comments (if provided):

Two reviewers have commented on the manuscript and made several pertinent points to improve the manuscript. Please address these point by point. The number of NDM resistant strains used is small and conclusions based on such small number should have a caveat.

Reviewers' comments:

Reviewer's Responses to Questions

**Comments to the Author**

1. Is the manuscript technically sound, and do the data support the conclusions?

Reviewer #1: Partly

Reviewer #2: Yes

2. Has the statistical analysis been performed appropriately and rigorously? 

Reviewer #1: No

Reviewer #2: Yes

3. Have the authors made all data underlying the findings in their manuscript fully available?

Reviewer #1: Yes

Reviewer #2: Yes

4. Is the manuscript presented in an intelligible fashion and written in standard English?

Reviewer #1: Yes

Reviewer #2: Yes

5. Review Comments to the Author

Reviewer #1: The manuscript entitled “Comparison of four low-cost carbapenemase detection tests and a proposal of an algorithm for early detection of carbapenemase-producing Enterobacteriaceae in resource-limited settings” describes the four different phenotypic methods (MHT, mCIM, CNPt, and Carba NP-direct) for carbapenemase detection and also proposed a protocol to simplify rapid detection of CP-CRE using the combination of CNPt-direct and mCIM. While the study brings information, required improvements, including statistical interpretation.

Major comments

Statistical interpretation of the results is not clearly mentioned in the manuscript, especially to compare among four low-cost carbapenemase detection tests that were used in this study.

Lines no 101-103: Details of the control microorganisms can be mentioned.

Line 163-164: Statistics analysis used for the study can be explained in detail, including the software used, if any.

Line 188 and table 2: The microorganisms used in this study, Enterobacteriaceae species, has not been isolated in this study. Hence, it can be mentioned with reference(s).

The proposed protocol for rapid detection of CP-CRE (Line 302) using the combination of CNPt-direct and mCIM, can be move to the result section, followed by discussion of the same in the discussion section.

Minor comments

Lines no: 51-52: Please rephrase the sentences

Line no 59: microorganisms, instead “organisms.”

Lines no 68-72: These two sentences are contrary to each other. The first/second sentence can be rephrased differently.

Line no 88: Expand the abbreviation, MHT, as appears for the first time

Line 93: Which “molecular methods” are authors talking about?

Lines 111, 122, 126, 137, 140, 153, 154: remove “ - ” between number and unit

Lines 210-222: This paragraph can be summarised.

Lines 221-222: This sentence can be removed from here

Lines 224-226: Include a reference

Line 226: ….we observed overall low sensitivity…….

Lines 227-228: Rephrase this sentence

Reviewer #2: Dear Authors

Th paper compares 4 techniques for detecting carbapenem resistance in CRE organisms. The paper compares the sensitivity, specificity, positive predictive value and negative predictive value for 4 methodologies as a comparison.

The paper brings out important problems in various methods for detecting carbapenamases. But the following can be added

1. What were the clinical sources of the isolates? The choice of antibiotic is also dependent on the site of infection hence this data could enhance the data in the paper

2. This is a pilot study kind of data as far as the numbers are concerned. For example, 4 NDM isolates were only present in the study. It would be important to explain that larger number of isolates may help in understanding the shortcomings of these tests better

3. When the authors mention Oxa-48 producers or NDM producers, how were these characterized? It would be important to also have this mentioned in the paper

4. Why were other CRO's like Pseudomonas or Acinetobacter not studied?

6. PLOS authors have the option to publish the peer review history of their article (what does this mean?). If published, this will include your full peer review and any attached files.

Reviewer #1: No

Reviewer #2: **Yes: **Dr. Anusha Rohit

---

## [Author Response · Author response to Decision Letter 0]

15 Dec 2020

 We have reformatted the manuscript according to the given PLOSOne guidelines.

2. Please provide more information on how another researcher could obtain the CRE strains used in this study so that they could reproduce your results.

Thank you for your suggestion. “All data are fully available without restriction” is already mentioned in the Data Availability section in the manuscript submission form. 

If further explanation is needed, following statement can be added.

Materials of the current study are available from the corresponding author upon a reasonable request.

3. To comply with PLOS ONE submission guidelines, in your Methods section, please provide additional information regarding your statistical analyses. In addition, please report your p-values to support your claims. For more information on PLOS ONE's expectations for statistical reporting, please see https://journals.plos.org/plosone/s/submission-guidelines.#loc-statistical-reporting.

Thank you for your comment. We have added additional information on statistical analysis in the Materials and Methods.

Additional Editor Comments (if provided):

Two reviewers have commented on the manuscript and made several pertinent points to improve the manuscript. Please address these point by point. The number of NDM resistant strains used is small and conclusions based on such small number should have a caveat.

We appreciate the reviewer’s constructive feedback on our manuscript.

We agree with the editor that our study contains a smaller number of NDM producers. We have added several sentences in the discussion section to highlight the limitations of our study.

“Although, this study consisted of a small sample of NDM producing CRE (n=4), these findings are in agreement with previous studies in which MHT has been found to have poor sensitivity (50 - 86%) for detection of MBLs (18, 28) and low specificity (91%) (18).” (Line no 244-246 in manuscript with track changes)

“Since the present study is mainly based on the OXA-48-like carbapenemase producing CRE, further studies with a large sample of CP-CRE isolates with adequate representation of commonly found genotypes (eg: KPC, NDM, OXA-48-like, IMP, VIM, etc.) may help better understanding of the performance of the proposed algorithm.” (Line no 330-334 in manuscript with track changes)

Reviewer #1: The manuscript entitled “Comparison of four low-cost carbapenemase detection tests and a proposal of an algorithm for early detection of carbapenemase-producing Enterobacteriaceae in resource-limited settings” describes the four different phenotypic methods (MHT, mCIM, CNPt, and Carba NP-direct) for carbapenemase detection and also proposed a protocol to simplify rapid detection of CP-CRE using the combination of CNPt-direct and mCIM. While the study brings information, required improvements, including statistical interpretation.

Major comments

Statistical interpretation of the results is not clearly mentioned in the manuscript, especially to compare among four low-cost carbapenemase detection tests that were used in this study.

Thank you for your comment. We have added additional information on statistical analysis in the Materials and Methods and have included 95% confidence intervals and P values in the results section.

Lines no 101-103: Details of the control microorganisms can be mentioned.

Thank you for the suggestion. We have added the details of the control organisms and have rewritten the section on bacterial isolates in the materials and methods.

Line 163-164: Statistics analysis used for the study can be explained in detail, including the software used, if any.

Thank you for your suggestion. We have added the relevant information.

Line 188 and table 2: The microorganisms used in this study, Enterobacteriaceae species, has not been isolated in this study. Hence, it can be mentioned with reference(s).

The proposed protocol for rapid detection of CP-CRE (Line 302) using the combination of CNPt-direct and mCIM, can be move to the result section, followed by discussion of the same in the discussion section.

Thank you for your comment. We have revised the Bacterial isolates in the methods section. Following sentence is mentioned with reference 20.

A total of 117 previously characterized Enterobacteriaceae clinical isolates were tested to evaluate the performance of four phenotypic carbapenemase detection assays, namely MHT, mCIM, CNPt, and CNPt-direct (20). 

Reference no 20: 

Kumudunie WGM, Wijesooriya LI, Namalie KD, Sunil-Chandra NP, Wijayasinghe YS. Epidemiology of multidrug-resistant Enterobacteriaceae in Sri Lanka: First evidence of blaKPC harboring Klebsiella pneumoniae. Journal of infection and public health. 2020;13(9):1330-5.

We think that there is some overlap between results and discussion sections. Therefore, we prefer to keep the proposed algorithm at the present location as it is the take home message of the manuscript.

Minor comments

Lines no: 51-52: Please rephrase the sentences

Thank you for your suggestion. We have rephrased the sentence.

Line no 59: microorganisms, instead “organisms.”

Replaced

Lines no 68-72: These two sentences are contrary to each other. The first/second sentence can be rephrased differently.

Thank you for pointing out. We have rephrased the second sentence as follow.

However due to high cost, unavailability of necessary instrumentation and technical expertise in many diagnostic laboratories, the PCR is not suitable for routine detection of CP-CRE in developing countries.

Line no 88: Expand the abbreviation, MHT, as appears for the first time

Thank you for pointing out. “modified Hodge test” is first appeared in Line 80 (Line 77 in original submission), so MHT was added next to it.

Line 93: Which “molecular methods” are authors talking about?

Thank you for pointing out. By “molecular methods” we broadly referred the techniques used to detect the presence of particular genes. To be more specific, we have replaced the “molecular method” with “PCR based detection methods”.

Lines 111, 122, 126, 137, 140, 153, 154: remove “ - ” between number and unit

Removed

Lines 210-222: This paragraph can be summarised.

Thank you for your suggestion. We have reduced the paragraph.

Lines 221-222: This sentence can be removed from here

Agreed. The sentence was removed.

Lines 224-226: Include a reference

Agreed. Two references (Ref 26 and 27) were added.

Line 226: ….we observed overall low sensitivity…….

This sentence was revised as,

we observed relatively low overall sensitivity (90.7%) and specificity (92.1%), and low positive predictive value (56.0%) for MHT (Table 1). 

Lines 227-228: Rephrase this sentence

Thank you for your suggestion. We have rephrased the sentence.

Reviewer #2: Dear Authors

Th paper compares 4 techniques for detecting carbapenem resistance in CRE organisms. The paper compares the sensitivity, specificity, positive predictive value and negative predictive value for 4 methodologies as a comparison.

The paper brings out important problems in various methods for detecting carbapenamases. But the following can be added

1. What were the clinical sources of the isolates? The choice of antibiotic is also dependent on the site of infection hence this data could enhance the data in the paper

Thank you for your comment. The information (sources, species and genetic identification, antibiotic susceptibilities, etc.) regarding the clinical isolates used in this study has been previously published and is available in reference 20 (an open access article). Reference 20 is cited in the materials and methods.

Reference no 20: 

Kumudunie WGM, Wijesooriya LI, Namalie KD, Sunil-Chandra NP, Wijayasinghe YS. Epidemiology of multidrug-resistant Enterobacteriaceae in Sri Lanka: First evidence of blaKPC harboring Klebsiella pneumoniae. Journal of infection and public health. 2020;13(9):1330-5.

2. This is a pilot study kind of data as far as the numbers are concerned. For example, 4 NDM isolates were only present in the study. It would be important to explain that larger number of isolates may help in understanding the shortcomings of these tests better

We agree that our study contained only a very low number of NDM isolates. Even with this low numbers, similar trend was observed when compared to the published studies. Therefore, the following sentences were added to the discussion section in the manuscript. 

“Although, this study consisted of a small sample of NDM producing CRE (n=4), these findings are in agreement with previous studies in which MHT has been found to have poor sensitivity (50 - 86%) for detection of MBLs (18, 28) and low specificity (91%) (18).” (Line no 244-246 in manuscript with track changes)

“Since the present study is mainly based on the OXA-48-like carbapenemase producing CRE, further studies with a large sample of CP-CRE isolates with adequate representation of commonly found genotypes (eg: KPC, NDM, OXA-48-like, IMP, VIM, etc.) may help better understanding of the performance of the proposed algorithm.” (Line no 330-334 in manuscript with track changes)

3. When the authors mention Oxa-48 producers or NDM producers, how were these characterized? It would be important to also have this mentioned in the paper

Thank you for the suggestion. We have revised the Bacterial isolates in the Materials and Methods with more details.

A total of 117 previously characterized Enterobacteriaceae clinical isolates were tested to evaluate the performance of four phenotypic carbapenemase detection assays, namely MHT, mCIM, CNPt, and CNPt-direct (20). These strains included 54 genetically identified CP-CRE producing carbapenemases of Ambler class A. B, and D: KPC (n=2), NDM (n=4), OXA-48-like (n=44), NDM+OXA-48-like (n=4). Three non-CP-CRE and 60 non-CRE consisted of 30 extended-spectrum β-lactamase producing Enterobacteriaceae (ESBL-PE: ESBL production was identified by double disc synergy test using 30 μg cefotaxime and 20/10 μg amoxicillin-clavulanic acid discs according to CLSI guidelines) and 30 non-ESBL-PE ( Enterobacteriaceae sensitive to both third generation cephalosporins and carbapenems) served as negative control in the study.

The detailed genetic characterization of the CP-CRE used in this study is available at 

Reference no 20: 

Kumudunie WGM, Wijesooriya LI, Namalie KD, Sunil-Chandra NP, Wijayasinghe YS. Epidemiology of multidrug-resistant Enterobacteriaceae in Sri Lanka: First evidence of blaKPC harboring Klebsiella pneumoniae. Journal of infection and public health. 2020;13(9):1330-5.

4. Why were other CRO's like Pseudomonas or Acinetobacter not studied?

Thank you for your comment. We understand that it is worthwhile to explore the carbapenem resistance in Pseudomonas and Acinetobacter as these CRO’s are also public health concerns. However, this manuscript is a product of a government funded, ongoing study on carbapenem resistant Enterobacteriaceae. The tests we studied in this research cannot be directly employed to detect other CRO’s. Therefore, the study of the carbapenem resistance in Pseudomonas and Acinetobacter would be the next step of our research.

---

## [Decision Letter · Decision Letter 1]

26 Dec 2020

Comparison of four low-cost carbapenemase detection tests and a proposal of an algorithm for early detection of carbapenemase-producing Enterobacteriaceae in resource-limited settings.

PONE-D-20-31489R1

Dear Dr. Wijayasinghe,

We’re pleased to inform you that your manuscript has been judged scientifically suitable for publication and will be formally accepted for publication once it meets all outstanding technical requirements.

Kind regards,

Iddya Karunasagar

Academic Editor

PLOS ONE

Additional Editor Comments (optional):

All reviewer comments have been addressed satisafactorily.

Reviewers' comments:

Reviewer's Responses to Questions

**Comments to the Author**

1. If the authors have adequately addressed your comments raised in a previous round of review and you feel that this manuscript is now acceptable for publication, you may indicate that here to bypass the “Comments to the Author” section, enter your conflict of interest statement in the “Confidential to Editor” section, and submit your "Accept" recommendation.

Reviewer #1: All comments have been addressed

2. Is the manuscript technically sound, and do the data support the conclusions?

Reviewer #1: (No Response)

3. Has the statistical analysis been performed appropriately and rigorously? 

Reviewer #1: (No Response)

4. Have the authors made all data underlying the findings in their manuscript fully available?

Reviewer #1: (No Response)

5. Is the manuscript presented in an intelligible fashion and written in standard English?

Reviewer #1: (No Response)

6. Review Comments to the Author

Reviewer #1: (No Response)

7. PLOS authors have the option to publish the peer review history of their article (what does this mean?). If published, this will include your full peer review and any attached files.

Reviewer #1: No

---

## [Editor Report · Acceptance letter]

4 Jan 2021

PONE-D-20-31489R1 

Comparison of four low-cost carbapenemase detection tests and a proposal of an algorithm for early detection of carbapenemase-producing *Enterobacteriaceae* in resource-limited settings 

Dear Dr. Wijayasinghe:

I'm pleased to inform you that your manuscript has been deemed suitable for publication in PLOS ONE. Congratulations! Your manuscript is now with our production department. 

Kind regards, 

on behalf of

Dr. Iddya Karunasagar 

Academic Editor

PLOS ONE